# OpenReview forum: "On Evaluating Adversarial Robustness of Large Vision-Language Models"
_NeurIPS.cc/2023/Conference — NeurIPS 2023 poster_

### Official Review · Reviewer_zQtG · 2023-06-11

**Soundness:** 3 good
**Presentation:** 3 good
**Contribution:** 3 good
**Rating:** 7
**Confidence:** 4

**Summary:**

This paper aims to assess the adversarial robustness of vision components in large vision-language models, which is an increasingly significant issue due to the prevalence of such models. The experiments conducted in this study are extensive, encompassing evaluations on UniDiffuser, BLIP, BILP-2, Img2Prompt, MiniGPT-4, and LLaVA. The research investigates transfer-based attacks and also employs black-box queries to enhance the effectiveness of the evaluation method.

**Strengths:**

The paper conducts extensive experiments on six prominent large vision-language models. By evaluating the adversarial robustness on these models, the study provides a comprehensive analysis that covers a wide range of architectures, ensuring the results are representative and applicable to the broader landscape of vision-language models.

One of the key strengths is that the paper demonstrates effective and realistic attacks specifically tailored for these models. By employing transfer-based attacks and utilizing black-box queries, the study goes beyond traditional methods to showcase the vulnerabilities and potential risks faced by large vision-language models in real-world scenarios.

**Weaknesses:**

This is an evaluation paper. It would be nice to see how baseline defense, such as adversarial training on CLIP [1], would help mitigate the adversarial vulnerability of the proposed attack.

The method focuses on the vision part, which is easier to attack, what about attacks on the language?

[1] Mao et al. Understanding Zero-Shot Adversarial Robustness for Large-Scale Models. ICLR 2023.

**Questions:**

What are the computational resources used to reproduce the paper?

**Limitations:**

None.

---

> ### Author Rebuttal · Authors · 2023-08-09
>
> Thank you for your supportive review and suggestions, we have uploaded a rebuttal PDF.
>
> ***Q1: It would be nice to see how baseline defense, such as adversarial training on CLIP [1], would help mitigate the adversarial vulnerability of the proposed attack***
>
> Thank you for the insightful comments. Following your suggestion, we apply the defense method such as adversarial training on CLIP [1], and evaluate our proposed attacking methods under the defense (we use the model checkpoints provided by [1]). The results are shown in **Table E** of the rebuttal PDF, where we employ the same experimental setups as in Table 2 of the main paper. As can be seen, even with the defense (i.e. adversarial training on CLIP by [1]), our attacking methods can achieve comparable performance to our original results, demonstrating the effectiveness of our attacking methods. These experiment results and discussion will be included in the revision.
>
> ***Q2: The method focuses on the vision part, which is easier to attack, what about attacks on the language?***
>
> When an adversary attempts to fool a VLM, it can choose to attack either the vision or the language part, or both. We show in our paper that simply attacking the vision part  is sufficient to fool the entire VLM into returning targeted responses. As a result, fooling a VLM by attacking the vision part would be a more effective strategy than attacking the language part. Still, attacking VLMs through languages would be intriguing directions to pursue, and we will include relevant discussion in the revision.
>
> ***Q3: What are the computational resources used to reproduce the paper?***
>
> In Table I of Section A (in the Appendix), we have included detailed computational resources used in each experiment (Figure/Table) of our work, including (1) the type of GPU we used, (2) GPU hours, and (3) carbon emission. On the other hand, we hope to highlight that our proposed method is computationally efficient and can run on a single GPU.
>
> ***References:*** \
> [1] Mao et al. Understanding Zero-Shot Adversarial Robustness for Large-Scale Models. ICLR 2023

---

> > ### Comment · Reviewer_zQtG · 2023-08-19
> > **A follow up question**
> >
> > The reviewer thanks the author, and the rebuttal answered my question 2,3.
> >
> > For Q1, I have a follow-up question, what adversarial variant does the experiment in Table E use? Is it the prompting or finetuning? Since the adversarial training is not almost not defending the attack, the reviewer is worried that the adversarial training experiment is not optimally conducted. More details should be provided, like what dataset the method finetuned on, how many epochs, and explanations on why the defense results are worse than the finding in [1].

---

> > > ### Author Response · Authors · 2023-08-20
> > > **Thank you for your feedback**
> > >
> > > Thank you for your feedback. We used the checkpoint corresponding to the adv-CLIP called `TeCoA w/ Finetuning’, as provided in https://github.com/cvlab-columbia/ZSRobust4FoundationModel (the official code released by [1]). In Table E, we applied adv-CLIP to defend our transfer-based MF-ii (i.e., using adv-CLIP as the surrogate model for MF-ii), and we left the clean-CLIP encoder in Unidiffuser unchanged because we observe that **substituting it with adv-CLIP will significantly degrade the clean performance of Unidiffuser**.
> > >
> > > Nevertheless, we focus on the robust performance in this response, and in Table F below, we report additional results of using adv-CLIP to defend our query-based MF-tt (i.e., substituting the clean-CLIP encoder in Unidiffuser with adv-CLIP), or defending both MF-ii and MF-tt.
> > >
> > > ***Tabel F: Results of attacking Unidiffuser***
> > > |     Method      | RN50 | RN101 | ViT-B/16 | ViT-B/32 | ViT-L/14 | Ensemble |
> > > |---------------|:-----------:|-----------:|-----------:|-----------:|-----------:|-----------:|
> > > | Clean image |   0.417   |   0.415   |   0.429   |  0.446   | 0.305 | 0.402 |
> > > | MF-ii (adv-CLIP) + MF-tt (adv-CLIP)   |   0.584   |   0.549   |   0.583   |  0.611   | 0.456 | 0.557 |
> > > | MF-ii (clean-CLIP) + MF-tt (adv-CLIP)   |   0.698   |   0.679   |   0.719   |  0.731   | 0.632 | 0.691 |
> > > | MF-ii (adv-CLIP) + MF-tt (clean-CLIP)   |   0.739   |   0.724   |   0.755   |  0.766   | 0.675 | 0.731 |
> > > | MF-ii (clean-CLIP) + MF-tt (clean-CLIP)   |   0.748   |   0.734   |   0.759   |  0.773   | 0.684 | 0.739 |
> > >
> > > Please note that for transfer-based MF-ii, **the adversary can freely choose its surrogate model (e.g., either clean-CLIP or adv-CLIP)**, so the results in Table F demonstrate that even if the encoder in Unidiffuser is enhanced by adv-CLIP (i.e., MF-tt (adv-CLIP)), our attacks can still apply clean-CLIP in MF-ii (i.e., MF-ii (clean-CLIP)) to conduct a good initialization and efficiently evade Unidiffuser by combining with MF-tt (i.e., **the results in the row of MF-ii (clean-CLIP) + MF-tt (adv-CLIP)**).
> > >
> > > The reason why adv-CLIP may be less effective at protecting large VLMs such as Unidiffuser is that the CLIP encoder only occupies a small portion of these VLMs (e.g., the CLIP encoder has 300M parameters, Unidiffuser has 1.4B parameters, and MiniGPT-4/LLaVA has more than 13B parameters). As a result, a strong defense may necessitate improving both the CLIP encoder and the language model in these VLMs. We will incorporate these discussions into the revision and conduct additional experiments on other large VLMs enhanced by [1].
> > >
> > > ***Reference:*** \
> > > [1] Mao et al. Understanding Zero-Shot Adversarial Robustness for Large-Scale Models. ICLR 2023

---

> > > > ### Comment · Reviewer_zQtG · 2023-08-20
> > > > **Thank you for your clarification**
> > > >
> > > > The reviewer thanks the authors' reply.
> > > >
> > > > After checking paper [1], the released model are trained for 1/255 robustness, while this work's attack is 8/255 which is a larger bound. This can explain the why defense is not effective under the proposed attack because they are not adversarially trained on 8/255 attack. The author ideally should run [1] with the proposed 8/255 bounded attack for adversarial training, which might defend some of the attack in vision model in this work. Or launch their attack with 1/255 against the VLM + CLIP model to reuse the checkpoint in [1].

---

> > > > > ### Author Response · Authors · 2023-08-20
> > > > > **Thank you for the further comments**
> > > > >
> > > > > Thank you for the further comments.
> > > > >
> > > > > We need to emphasize that, because we aim to fool large VLMs into returning *targeted* responses under *black-box* settings, we cannot achieve this difficult goal with a perturbation budget of 1/255, as ablated in Figure 6 of the main paper.
> > > > >
> > > > > We would like to evaluate the clean and robust performance of enhancing large VLMs with adv-CLIP under the 8/255 threat model, but because the authors of [1] only released 1/255 checkpoint, we are unable to meet the rebuttal deadline (only 1 day left) to retrain adv-CLIP models under the 8/255 threat model (we note that there are no results in [1] for the 8/255 threat model, so we must communicate with the authors to ensure that we correctly reimplement their model for fair comparison). We will conduct these experiments and report the full results in the revision.

---

> > > > > > ### Comment · Reviewer_zQtG · 2023-08-20
> > > > > > **The authors addressed my questions**
> > > > > >
> > > > > > Thank you for clarification. The plan makes sense.

---

> > > > > > > ### Author Response · Authors · 2023-08-20
> > > > > > > **Thank you**
> > > > > > >
> > > > > > > Thank you for the insightful discussion and raising the rating. :-)

---

### Official Review · Reviewer_UbWE · 2023-06-15

**Soundness:** 3 good
**Presentation:** 4 excellent
**Contribution:** 3 good
**Rating:** 6
**Confidence:** 4

**Summary:**

The paper evaluates the pixel-space adversarial robustness of large vision-language models (VLMs), where the targeted attack has only black-box access to the large VLM systems. The paper introduces two adversarial strategies: transfer-based and query-based. The transfer-based strategy performs white-box attacks on surrogate models like CLIP and BLIP and transfers to large VLMs including MiniGPT-4 and LLaVA. The query-based strategy uses transfer-based priors to improve the efficacy of targeted evasion against these VLMs. The authors demonstrate the effectiveness of these strategies through comprehensive evaluations.



**Strengths:**

1. The paper is well-presented with good motivation, and the authors provide a clear storyline with visual demonstrations to validate the proposed approach.

2. It is novel to optimize the adversarial objective by matching image-image features (MF-ii) on large VLMs. The following Query-based attacking strategy (MF-tt) further enhances the effectiveness of the attack.

3. The authors analyze the approach from multiple perspectives. The paper includes discussions on the influences of the perturbation, and the Grad-CAM interpretation visualizes the model decision well.

**Weaknesses:**

1. The black-box setting is questionable. Although the adversary only has black-box access to the victim system $p_{\theta}$, the transfer-based attack has white-box access to foundation components like CLIP. There is a high chance that the victim system shares mutual information (e.g., $p_{\theta}$ contains CLIP / $p_{\theta}$ trains on the same vision-language dataset / $p_{\theta}$ has similar model architecture) with these white-box components. It will be beneficial if the authors elaborate more on the source of transferability.

2. Although the Query-based attacking strategy (MF-tt) is feasible, the cost of such a process can be high. What is the forward inference cost (e.g., time of API call) per query, and how many queries are needed per attack? Authors should show the cost effectiveness of query attacks.

3. The experiments mainly focus on imposing MF-ii as a prior for the following MF-tt queries. It will be constructive If authors demonstrate more on the sole use of MF-tt (which aligns with the black-box setting and is more computationally lightweight) without a strong MF-ii prior for the query attack.

**Questions:**

1. The authors give Section 4.3 the title *VLMs induce semantic adversarial perturbations*. However, what Fig. 5 shows is not semantic adversarial examples but inference results of pixel-wise adversarial noises. The term semantic adversarial perturbation commonly indicates semantic variations that fail the model [1]. Is Fig. 5 trying to discuss *does VLM adversarial perturbations induce semantic meanings*?

2. Since the pixel-space adversarial noises are not intuitive/interpretable to human observers, are there any future discussions on generating VLM attacks based on physically-observable attributes or languages [1,2,3,4]?

3. To conclude, this paper is a timely contribution to the field. I will consider revising the rating based on the soundness of the authors' responses.

[1] ICCV 2019, Semantic Adversarial Attacks: Parametric Transformations That Fool Deep Classifiers.

[2] ECCV 2020, Semanticadv: Generating adversarial examples via attribute-conditioned image editing

[3] CVPR 2023, Zero-shot Model Diagnosis

[4] ICLR 2023, Diagnosing and Rectifying Vision Models using Language

**Limitations:**

Yes, the authors have stated the potential limitations of the work.

---

> ### Author Rebuttal · Authors · 2023-08-09
>
> Thank you for your supportive review and suggestions, we have uploaded a rebuttal PDF.
>
> ***Q1: There is a high chance that the victim system shares mutual information with these white-box components. It will be beneficial if the authors elaborate more on the source of transferability***
>
> Indeed, existing open-source large VLMs are primarily composed of publicly available modules (e.g., CLIP and Vicuna) and are finetuned on some common vision-language datasets. These facts increase the chance that the surrogate model used by black-box adversaries shares mutual information with the victim model, essentially making the victim model vulnerable to adversarial transferability.
>
> Nevertheless, we want to emphasize two points to demonstrate the significance of our transfer-based attacks. First, even if $p\_{\\theta}$ contains CLIP, the CLIP module only occupies a small portion of the large VLMs (e.g., a CLIP encoder has 300M parameters and LLaVA/MiniGPT-4 has 13.3B/14.1B parameters), where the main model capacity is involved in large language models that are unseen to our transfer-based attackers. Second, while knowing the CLIP module may be sufficient for *untargeted* attacks against large VLMs, we focus on much more difficult *targeted* settings, and it is non-trivial to fool large VLMs to return targeted responses solely based on adversarial transferability. On the other hand, as discussed in our response to **Q3**, we demonstrate that using query-based MF-tt alone is still effective against various VLMs, without transfer-based MF-ii as the prior.
>
> ***Q2: What is the forward inference cost (e.g., time of API call) per query, and how many queries are needed per attack?***
>
> As stated in Line 176, we set query times $N=100$ and update the adversarial images by 8-steps PGD for MF-tt, resulting in a total query time of $800$ per attack in our setup. In **Table C** of the rebuttal PDF, we provide the time of API call per query for different large VLMs used in our experiments. We will include more details in the revision.
>
> ***Q3: It will be constructive If authors demonstrate more on the sole use of MF-tt (which aligns with the black-box setting and is more computationally lightweight).***
>
> In **Table D** of the rebuttal PDF, we show the experiment results of using only MF-tt over various VLMs (e.g., Unidiffuser, BLIP-2, LLaVA). Our results indicate that our query-based attack MF-tt is effective against various VLMs. However, when MF-ii is used as the prior, it can achieve a better performance. More results will be provided in the revision.
>
> ***Q4: Is Fig. 5 trying to discuss whether VLM adversarial perturbations induce semantic meanings?***
>
> Thank you for pointing this out. Yes, in Section 4.3 and Fig. 5, we attempt to discuss whether the crafted adversarial perturbations induce any semantic meanings. We will correct the claims in the revision.
>
> ***Q5: Are there any future discussions on generating VLM attacks based on physically-observable attributes or languages?***
>
> Thank you for the insightful comments. In our paper, we constrain the pixel perturbations to be small (e.g., $\\epsilon=8$) enough to be visually imperceptible (or difficult to perceive) by humans, and we show that simply attacking the visual modality is sufficient to fool the entire VLMs. Nonetheless, attacking VLMs through physically-observable attributes on the vision part (e.g., adding eyeglasses or other accessories to the human portrait [1], modifying hair color or style [2, 3]) or languages [4] would be promising directions to pursue, and we will include relevant discussion on these works in the revision.
>
>
> ***References:*** \
> [1] Joshi et al. Semantic Adversarial Attacks: Parametric Transformations That Fool Deep Classifiers. ICCV 2019 \
> [2] Qiu et al. SemanticAdv: Generating Adversarial Examples via Attribute-conditional Image Editing. ECCV 2020 \
> [3] Luo et al. Zero-shot Model Diagnosis. CVPR 2023 \
> [4] Zhang et al. Diagnosing and Rectifying Vision Models using Language. ICLR 2023

---

> > ### Comment · Reviewer_UbWE · 2023-08-11
> >
> > I appreciate the revision and the insights provided by the rebuttal. The experiments are sufficiently good. I will raise the rating to 6.

---

> > > ### Author Response · Authors · 2023-08-12
> > > **Thank you for your feedback**
> > >
> > > We appreciate your detailed comments and suggestions. We will polish our paper further and incorporate new results into the final revision. Thank you again!

---

### Official Review · Reviewer_Xvuc · 2023-07-02

**Soundness:** 3 good
**Presentation:** 3 good
**Contribution:** 3 good
**Rating:** 7
**Confidence:** 4

**Summary:**

This paper focuses on black-box targeted adversarial attacks on multimodal vision/language models via transfer. They observe particular vulnerability to transfer attacks because an adversarial image can be constructed in a fully-differentiable manner w.r.t. a model like CLIP and then transferred over to the black box target model. Their final approach involves matching the adversarial image features to both text and image features in the surrogate models used for training.

**Strengths:**

I think the combined method MF-it and MF-ii is clever, and I think the results are impressive.

I find the paper clearly valuable and timely since models with these vulnerabilities are currently being deployed and their vulnerabilities could pose risks. I appreciate the discussion of this in the conclusion.

I think that the experiments are compelling. I do not spot errors.

**Weaknesses:**

(Minor) I find the figs to be generally cluttered.

I would have liked to see experiments related to real-world concerns such as ones that were meant to illustrate real ways that these attacks could cause problems. For example, evading NSFW filters or making language models provide misinformation about an image that is used in a query.

**Questions:**

Could these vulnerabilities have been foreseen (probably yes)? Have they been addressed at all by the people open-sourcing VLMs? What are the most immediate risks posed by deployed VLMs? What does this mean for AI norms and governance? I would welcome more discussion of these in the discussion section.

**Limitations:**

This paper did not focus on defenses for these attacks, but I would not have expected it to.

The use of GradCAM to analyze the attacks is limited and based on visual inspection. Also GradCAM and similar methods are well known to have many problems. Given my familiarity with some of the problems with feature attribution methods, these experiments don't really add much to the paper in my opinion.

---

> ### Author Rebuttal · Authors · 2023-08-09
>
> Thank you for your supportive review and suggestions, we have uploaded a rebuttal PDF.
>
> ***Q1: (Minor) I find the figs to be generally cluttered***
>
> Thank you for the comments. In the revision, we will rearrange our figures to make the content more clear.
>
> ***Q2: I would have liked to see experiments related to real-world concerns. For example, evading NSFW filters or making language models provide misinformation about an image that is used in a query***
>
> In this work, we used MiniGPT-4, LLaVA, and Img2Prompt to evaluate the visual question-answering (VQA) task, in which an image is indeed used in a query and the adversarial goal is to make language models provide misinformation. The results have been demonstrated in Figure 3 (main paper), Figures IV and V (Appendix). These results are highly relevant to real-world concerns, because we can fool the model into returning *targeted* outputs, which could contain any predefined misinformation or bypass NSFW filters.
>
> ***Q3: Could these vulnerabilities have been foreseen (probably yes)? Have they been addressed at all by the people open-sourcing VLMs?***
>
> These vulnerabilities were most likely foreseen by OpenAI, and we believe this is one of the reasons why OpenAI is delaying the release of GPT-4's visual input. Unfortunately, our results indicate that these vulnerabilities are rarely addressed or even paid attention by the people open-sourcing VLMs. Existing open-source VLMs still primarily compete for the performance on standard benchmarks, while these VLMs are mostly made up of publicly accessible modules that do not take adversarial robustness into consideration.
>
> ***Q4: What are the most immediate risks posed by deployed VLMs? What does this mean for AI norms and governance? I would welcome more discussion of these.***
>
> Thank you for the questions. There have been attempts to deploy large VLMs into robots or autonomous driving systems, where the visual inputs could be the images captured by a camera (i.e., interaction with environments) and the language inputs could be user commands (i.e., interaction with humans). In these scenarios, adversarial attacks on VLMs could mislead robots or autonomous driving systems, inducing them to engage in risky behaviors. Improved adversarial robustness of VLMs and more general multi-modal models are thus critical for AI norms and governance.
>
> ***Q5: This paper did not focus on defenses for these attacks, but I would not have expected it to***
>
> Following your suggestion, we apply a recent defense method that adversarially trains a CLIP model [1], and evaluate our proposed attacking methods under the defense (we use the model checkpoints provided by [1]). The results are shown in **Table E** of the rebuttal PDF. As can be seen, even with the defense method by [1], we still can achieve comparable performance to our original results, demonstrating the effectiveness of our attacking methods. These experiment results and discussion will be included in the revision.
>
> ***Q6: The use of GradCAM to analyze the attacks is limited and based on visual inspection. Also GradCAM and similar methods are well known to have many problems***
>
> Thank you for your constructive feedback; we will include your comments in the paper to clarify these limitations in the revision.
>
> ***References:*** \
> [1] Mao et al. Understanding Zero-Shot Adversarial Robustness for Large-Scale Models. ICLR 2023

---

> > ### Comment · Reviewer_Xvuc · 2023-08-10
> > **Reply**
> >
> > Thank you for the updates and comments. I have few further comments to discuss. I think my rating will most likely stay at a 6 but may increase to a 7. I will be paying attention to the rebuttal process and other reviews in the next week.
> >
> > What if any changes are being made to the paper regarding 3 and 4? Could the authors paste the updates in response to 6 into a reply?

---

> > > ### Author Response · Authors · 2023-08-11
> > > **Thank you for your feedback**
> > >
> > > Thank you for your timely feedback.
> > >
> > > ***Q (a): What if any changes are being made to the paper regarding 3 and 4?***
> > >
> > > Since an additional content page is allowed for the final revision of NeurIPS (if our paper is accepted), we will add additional subsections or paragraphs in Section 5 to involve the contents we responded to *Q3* and *Q4*. These contents include discussions such as adversarial vulnerabilities that could be foreseen but have not yet been addressed by the people open-sourcing VLMs; as well as AI norms, governance and the immediate risks posed by deployed VLMs.
> > >
> > > (We are unsure whether we understand this question correctly. Please let us know if there are any additional details you'd like to discuss.)
> > >
> > >
> > > ***Q (b): Could the authors paste the updates in response to 6 into a reply?***
> > >
> > > We will include more discussions about the motivation for visual inspection and the limitations of using GradCAM in Section 4.3 (after Lines 221-226), as summarized below:
> > >
> > > As to the motivation, we followed Img2Prompt [1] and used GradCAM to compute and visualize the relevancy score of image patches with respect to the question (i.e., `query’), as demonstrated in our interpreting analyses (Figure 7 in the main paper and Figure VI in the Appendix). We clarify that this visual inspection only provide an *intuitive* way to interpret our attacking mechanism.
> > >
> > > Nonetheless, we emphasize that the use of GradCAM as a feature attribution method has limitations [2]. GradCAM's inability to localize multiple occurrences of an object in an image has been observed. While multiple occurrences of the same object in an image are common, GradCAM causes inaccurate localisation of a heatmap for an object (due to the global average pooling operation). On the other hand, the continuous upsampling and downsampling processes may also result in signal loss prior to visualization.
> > >
> > >
> > > ***References:*** \
> > > [1] Guo et al. From Images to Textual Prompts: Zero-shot Visual Question Answering with Frozen Large Language Models. CVPR 2023 \
> > > [2] Chattopadhay et al. Grad-Cam++: Generalized Gradient-Based Visual Explanations for Deep Convolutional Networks. WACV  2018

---

> > > > ### Comment · Reviewer_Xvuc · 2023-08-20
> > > > **Raising my rating to a 7**
> > > >
> > > > I will move from a 6 to a 7

---

> > > > > ### Author Response · Authors · 2023-08-20
> > > > > **Thank you for raising the rating**
> > > > >
> > > > > Thank you for your detailed comments, we will incorporate these discussions and changes in the final revision. We also appreciate your decision to raise the rating to a 7, and if possible, *could you please also edit the rating in the original review?* :-) Thank you again!

---

### Official Review · Reviewer_csPu · 2023-07-09

**Soundness:** 4 excellent
**Presentation:** 4 excellent
**Contribution:** 3 good
**Rating:** 7
**Confidence:** 4

**Summary:**

The authors propose to generate adversarial attacks on different vision-language models (VLM) like BLIP, MiniGPT4, and UniDiffuser. The proposed method is simple and straightforward. To perturb the image, the authors propose to maximise the similarity (inner dot product) between the image-image / text-text / image-text features of CLIP. The attacked image is then transferred to the black box VLM models, which generates some text on these images. The similarity between the generated and the target text is calculated using unnormalised cosine similarity on the CLIP text encoder outputs. Maximising the similarity between the CLIP encodings of clean and target text and CLIP encodings of the clean and generated images of target text doesn't require any gradient approximation. But maximising the similarity between the target text encodings and generated text requires zero-order optimisation to get the gradients on the image. The authors demonstrate that the proposed attack completely fools the VLM models to generate text very similar to the target text.

**Strengths:**

* The proposed method is incredibly simple and, therefore highly valuable for the community
* The proposed method evaluates different types of VLM models, which makes the claims of the proposed method stronger.
* The problem identified is timely and highlights the importance of building robust VLM models in future.


**Weaknesses:**

* The proposed method shows the results of the PGD-100 attack, while in most of the black box attacks, it is observed that stronger attacks can overfit to the surrogate model resulting in worse transferability. Therefore, it would be good if the authors can also share the results on weaker attacks like PGD-10 and FGSM.

* I think comparing the text quality using the CLIP score may not give the true picture. I would be nice if the authors can try some other metrics to compare the text quality.

* For some of the approaches like LLaVA and MiniGPT-4 in Table-2, the results obtained using the proposed attacks like MF-it and MP-ii are very close (almost similar) to the clean image. Does this mean that the model is not able to get fooled? It would be nice if the authors can present results using some metrics which can quantify whether the model is fooled or not. This will help in getting a deeper understanding of the proposed method.


**Questions:**

It would be great if the authors can address the raised concerns in the weakness section.

**Limitations:**

The authors have properly addressed the limitations and negative societal impact of their work.

---

> ### Author Rebuttal · Authors · 2023-08-09
>
> Thank you for your supportive review and suggestions, we have uploaded a rebuttal PDF.
>
> ***Q1: It would be good if the authors can also share the results on weaker attacks like PGD-10 and FGSM.***
>
> In **Table A** of the rebuttal PDF, we follow the Reviewer’s suggestion to report additional results using fewer PGD steps (e.g., 10-steps and 50-steps) against different large VLMs (e.g., Unidiffuser, BLIP-2, and LLaVA). As can be seen, PGD-100 outperforms PGD-50 and PGD-10, though the benefits tend to saturate as more PGD steps are added. This indicates that PGD-100 is still not overfitting to surrogate models, which could be attributed to the complexity of large VLMs versus traditional image classification models.
>
> ***Q2: It would be nice if the authors can try some other metrics to compare the text quality.***
>
> Following your suggestion, we compare the text quality by computing the embedding similarity between the generated response and the targeted text using FlagEmbedding [1], a state-of-the-art text embedding model (ranked first on MTEB and C-MTEB benchmarks). The findings are presented in **Table B** of the rebuttal PDF, where we employ the same experimental setups as in Table 2 of the main paper. As demonstrated, using FlagEmbedding yields similar results to using the CLIP score, and we will add these results in the revision.
>
> Furthermore, we would like to clarify that in the main paper, we used various text encoders (ResNet-50/101, ViT-B/16, ViT-B/32, and ViT-L/14) to ensure a thorough assessment of text quality, and we also provide multiple visualized demonstrations (Figures 1, 2, 3 in the main paper and Figures II, III, IV, V in the Appendix) for human evaluation.
>
>
> ***Q3: For some of the approaches like LLaVA and MiniGPT-4 in Table 2, the results obtained using MF-it and MP-ii are very close to the clean image. Does this mean that the model is not able to get fooled?***
>
> Indeed, we observe that models with larger capacities (13.3B parameters for LLaVA and 14.1B parameters for MiniGPT-4) are more difficult to get fooled using only transfer-based attacks like MF-it and MF-ii, compared to those with smaller capacities such as BLIP and UniDiffuser. This could be because LLaVA and MiniGPT-4 both use a large language model (Vicuna-13B [2]) as the text generator, which is relatively robust to transfer-based attacks. Therefore, we use transfer-based MF-ii to initialize the adversarial perturbation and query-based MF-tt to further tune the perturbation (see Figure 4 for details), which is effective as shown in Table 2 and Figure 3 (more results can be found in the Appendix).
>
>
> ***Q4: It would be nice if the authors can present results using some metrics which can quantify whether the model is fooled or not***
>
> We additionally conduct user studies to determine whether the model is fooled or not. We specifically categorize the generated responses into four types, which are as follows: *Type-(1)*: the model is completely fooled, i.e., it can generate the targeted response; *Type-(2)*: the model is partially fooled and can generate concepts/information related to the target text/image; *Type-(3)*: the model is partially fooled and can generate non-targeted response that is neither related to the target text/image nor the original clean image; *Type-(4)*: the model cannot be fooled, We randomly selected 100 VLM responses and asked users to vote on whether these models were fooled. The results are as follows:
>
> |     VLM      | Type-(1) | Type-(2) | Type-(3) | Type-(4) |
> |---------------|:-----------:|-----------:|-----------:|-----------:|
> | Unidiffuser |   84.5\%   |   8\%   |   5\%   |   2.5\%   |
> | BLIP-2       |   86\%   |   9.5\%   |   2.5\%   |   2\%   |
>
> As shown in our user studies, these VLMs can be fooled to a large extent in most cases, similar to the visualization results in our figures. Due to time constraints, we only evaluate Unidiffuser and BLIP-2 here, but we will conduct user studies on other large VLMs in the revision.
>
> ***References:*** \
> [1] https://github.com/FlagOpen/FlagEmbedding \
> [2] Chiang, et al. Vicuna: An open-source chatbot impressing gpt-4 with 90%* chatgpt quality.

---

> > ### Comment · Reviewer_csPu · 2023-08-19
> >
> > I want to thank the authors for providing a comprehensive rebuttal. I appreciate the authors for accepting that their method is difficult to apply for larger VLMs. I think this work shows a very interesting direction and would be valuable for the community. I will keep up my score.

---

> > > ### Author Response · Authors · 2023-08-19
> > > **Thank you for your feedback**
> > >
> > > We appreciate your detailed comments and suggestions. We will polish our paper further and incorporate new results into the final revision. Thank you again!

---

### Author Rebuttal · Authors · 2023-08-09

We thank all reviewers for their constructive feedback, and we have responded to each reviewer individually. We have also uploaded a rebuttal PDF that includes:

- **Table A**: CLIP score ($\\uparrow$) with fewer PGD steps against different VLMs;
- **Table B**: FlagEmbedding score ($\\uparrow$) as additional text quality metric against different VLMs;
- **Table C**: The average time of API call of each query of  different VLMs;
- **Table D**: CLIP score ($\\uparrow$) with the sole use of MF-tt against  different VLMs;
- **Table E**: CLIP score ($\\uparrow$) with the defense method applied (Mao et al. ICLR 2023).

---

### Decision · Program_Chairs · 2023-09-21

**Decision:**

Accept (poster)

**Comment:**

This paper evaluates the pixel-space adversarial robustness of large vision-language models (VLMs), where the targeted attack has only black-box access to the large VLM systems. The authors have done a nice job of rebuttal. After rebuttal, this paper received scores of 6777. Most reviewers are confident about their judgement. Specifically, the reviewers have commented that (1) the proposed method is simple and the studied problem is valuable for the community; (2) the experiments are compelling and comprehensive; (3) the paper is well-presented with good motivation. Therefore, the AC decided to recommend acceptance of the paper.